# Docosahexaenoic Acid Inhibits Proliferation of EoL-1 Leukemia Cells and Induces Cell Cycle Arrest and Cell Differentiation

**DOI:** 10.3390/nu11030574

**Published:** 2019-03-07

**Authors:** Kalliopi Moustaka, Eirini Maleskou, Andromachi Lambrianidou, Stelios Papadopoulos, Marilena E. Lekka, Theoni Trangas, Eirini Kitsiouli

**Affiliations:** 1Laboratory of Biochemistry, Department of Biological Applications & Technologies, University of Ioannina, 45110 Ioannina, Greece; kallia.moustaka@gmail.com (K.M.); emaleskou1988@gmail.com (E.M.); mahilabrianidou@hotmail.com (A.L.); ttrangas@cc.uoi.gr (T.T.); 2Laboratory of Biochemistry, Department of Chemistry, University of Ioannina, 45110 Ioannina, Greece; stelios_ppp@yahoo.com (S.P.); mlekka@uoi.gr (M.E.L.)

**Keywords:** Docosahexaenoic, EoL-1, G0/1 cycle arrest, differentiation, *PTAFR*, *PLA2G4A*, *MYC*

## Abstract

Τhe effect of docosahexaenoic acid (DHA, an omega-3 polyunsaturated fatty acid) upon the proliferation of EoL-1 (Eosinophilic leukemia) cell line was assessed, while additional cellular events during the antiproliferative action were recorded. DHA inhibited EoL-1 cells growth dose-dependently by inducing growth arrest at G0/1 phase of the cell cycle. After DHA addition to the cells, the expression of *MYC* oncogene was decreased, *PTAFR*-mRNA overexpression was observed which was used as a marker of differentiation, and *PLA2G4A*-mRNA increase was recorded. The enzymatic activities of phospholipase A_2_ (PLA_2_), a group of hydrolytic enzymes, whose action precedes and leads to PAF biosynthesis through the remodeling pathway, as well as platelet activating factor acetylhydrolase (PAFAH) which hydrolyses and deactivates PAF, were also measured. DHA had an effect on the levels of both the intracellular and secreted activities of PLA_2_ and PAFAH. The inflammatory cytokines IL-6 and TNF-α were also detected in high levels. In conclusion, DHA-induced EoL-1 cells differentiation was correlated with downregulation of *MYC* oncogene, overexpression of *PTAFR* and *PLA2G4A*-mRNAs, increase of the inflammatory cytokines production, and alteration of the enzymatic activities that regulate PAF levels. DHA is a natural substance and the understanding of its action on EoL-1 cells on molecular level could be useful in further investigation as a future therapeutic tool against F/P ^+^ hypereosinophilic syndrome.

## 1. Introduction

Docosahexaenoic acid (DHA) is a highly polyunsaturated omega-3 fatty acid (ω-3 or n-3). Omega-3 polyunsaturated fatty acids (PUFAs) are essential fatty acids which must be obtained from nutrition. Currently, the western diet contains a high ratio of n-6 PUFAs/n-3 PUFAs amount, and this is thought to contribute to cardiovascular disease, inflammation, and cancer [1].

In particular, omega-3 fatty acids (FAs) are considered necessary for human health because, esterified in phospholipids, they participate in cell membranes structure, they affect the function of cell membrane receptors and regulate gene function [2]. Furthermore, they are precursors of hormones [3] regulating blood clotting, contraction, and relaxation of artery walls, and inflammation, among others. Due to these effects, omega-3 fats have been shown to help prevent heart disease and stroke, resolve inflammation [4], and may play protective roles in cancer [5] and other diseases.

The main omega-3 fatty acids are docosahexaenoic (DHA, 22:6), eicosapentaenoic (EPA, 20:5), and alpha-linolenic (ALA). In the human body, the conversion of ALA into EPA and DHA is very limited [6]. DHA [(4Z,7Z,10Z,13Z,16Z,19Z)-docosa-4,7,10,13,16,19-hexaenoic acid, C_22_H_32_O_2_] is a very long-chain fatty acid with a 22-carbon backbone and 6 double bonds, originating from the 3rd, 6th, 9th, 12th, 15th, and 18th positions from the methyl end [7]. Chemical dynamic modeling studies support the idea that DHA and EPA contribute in the substantial increase of cell membranes fluidity. The DHA molecule possesses unique properties due to its high length and unsaturation, therefore, its effects on influencing cell signaling, protein trafficking, and cell cytokinetics [6] and controlling cell biochemistry and physiology [3] are under investigation.

The EoL-1 cell line is a cell type of acute myeloid (eosinophilic) leukemia. The origin was established at diagnosis from the peripheral blood of a 33-year-old man with acute myeloid (eosinophilic) leukemia following hypereosinophilic syndrome and is described to carry the fusion *FIP1L1-PDGFRα* gene [8]. The EoL-1 cell line is used as an in vitro model for the study of FIP1L1-PDGFRA-positive chronic eosinophilic leukemia [9] and is particularly useful for analyzing leukemic cell differentiation and the properties of malignant eosinophils [10]. EoL-1 cells differentiate not only phenotypically but also functionally into eosinophils by a number of stimuli (including alkaline pH, dimethylsulfoxide, TNF-α, G-CSF + TNF-α, HIL-3-derived factor, dibutyryl cAMP, IFN-γ). Furthermore, the differentiation of EoL-1 cells by cytokines, is associated with downregulation of *MYC* oncogene expression [11].

EoL-1 cells differentiate into eosinophils in the presence of n-butyrate, a very short chain fatty acid. Generally, n-butyrate decreases the proliferation of EoL-1 cells, without attenuating the level of *FIP1L1-PDGFRα* mRNA, by inhibiting nuclear deacetylases which results in the hyperacetylation of histones, to altered gene transcription and differentiation [12], while it induces the expression of markers for mature eosinophils [13]. The differentiation of EoL-1 cell line by n-butyrate is also associated with the induction of platelet activating factor receptor (*PTAFR*) gene expression, as indicated by *PTAFR*-mRNA accumulation [14]. Undifferentiated EoL-1 cells do not produce platelet activating factor (PAF), however their differentiation by IFNγ- restores the ability of PAF production in response to Ca^2+^ionophore. Interestingly, DHA supplementation into the culture of IFNγ-differentiated EoL-1 cells causes a decrease in PAF production [15].

Platelet activating factor (1-*O*-alkyl-2-acetyl-*sn*-glycero-3-phosphocholine) is a potent inflammatory mediator and an eosinophil chemotactic factor, which not only induces chemotaxis but also enhances many inflammatory functions of eosinophils such as release of eosinophil peroxidase and other granular contents from human eosinophils. PAF acts via binding to PAF-receptor and stimulates numerous signal transduction pathways including phospholipase A_2_, C, D, mitogen-activated protein kinases (MAPKs), and the phosphatidylinositol-calcium second messenger system [16]. Therefore, the PAF/*PTAFR* pathway of inflammation is considered as an active signaling route in normal, mature eosinophils.

Many studies have shown that docosahexaenoic acid exhibits a time- and concentration-dependent antiproliferative effect on various human cancer cell lines while having minimal cytotoxicity on the normal or non-tumorigenic cells [5,17], cause cell cycle arrest, or even apoptosis and presents synergistic anticancer properties with other drug substances [1,18,19]. Enormous data from cancer cell lines and in vivo cancer models have given insight into the mechanisms underlying the anticancer effects of ω-3 PUFAs [20,21].

In the present study, we investigated the antiproliferative and differentiating effects of DHA on EoL-1 cells. *PTAFR*-mRNA expression was measured as a marker of differentiation. Following DHA-treatment on EoL-1 cells, we also recorded additional molecular events, regarding phospholipases A_2_, trying to address the mode of function of DHA.

## 2. Materials and Methods

### 2.1. Cell Culture and Proliferation Assay

The EoL-1 cell line was provided by Genotype SA (Athens, Greece). The cells were cultured in RPMI 1640 medium (Biochrome KG, GIBCO 31870, Co Dublin, Ireland) supplemented with 10% heat-inactivated fetal bovine serum (Fetal Bovine Serum-FBS, 10270, GIBCO, Co Dublin, Ireland), 2 mM glutamine (L-Glutamine 200 mM, 25030, GIBCO, Co Dublin, Ireland), penicillin 100 units/mL (Penicillin G sodium 10,000 units/mL, GIBCO, Co Dublin, Ireland) and streptomycin 100 mg/mL (Streptomycin Sulfate 10,000 μg/mL and 25 μg/mL amphotericin B as fungizone, GIBCO, Co Dublin, Ireland) at 37 °C in a humidified atmosphere of 5% carbon dioxide (SANYO CO_2_, INCUBATOR, LabX, Midland, ON, Canada). The above complete medium was renewed every 3–4 days.

The cell number was adjusted to 1 × 10^6^ cells/mL with fresh complete medium. The viability of the cells was measured by Trypan Blue exclusion test (Blue solution 0.4% in NaCl, SIGMA, 0.2% final in PBS, phosphate buffer saline, GIBCO, Co Dublin, Ireland) every 3 days by cell counting on a Neubauer hemocytometer with a microscope (Kruss, Hamburg, Germany), as well as by MTT assay (CellTiter 96^®^ AQueous One Solution Cell Proliferation Assay, Promega Corporation, Madison, WI, USA).

### 2.2. Cell Treatment

Cell splitting was performed as follows: the content of the flask was transferred into a falcon plastic tube (Corning^®^ PP Centrifuge Tubes, with Plug Seal Cap, Sterile, Merck KGaA, Darmstadt, Germany) and centrifuged at 500× *g* (ROTOFIX 32, Andreas Hettich GmbH & Co. KG, Tuttlingen, Germany) for 10 min. The supernatant was discarded and the cell pellet was resuspended with complete medium. Cell counting was performed by the method of Trypan Blue staining.

For studying the effect of DHA on cell proliferation, EoL-1 cells were suspended at a concentration of 1 × 10^6^ cells/mL in complete medium containing different concentrations of DHA or 500 μM butyrate. Two DHA (cis-4,7,10,13,16,19-docosahexaenoic acid, minimum 98%, D 2534, SIGMA) concentrations of 30 mM and 3 mM in ethanol were used to adjust the range of concentrations of DHA.

The DHA solutions were stored at −20 °C, whereas during their use, they were kept in ice to avoid ethanol evaporation. The control cells were treated with the same amount of vehicle alone. The final ethanol concentration never exceeded 0.17% (*v*/*v*). Among the 10, 20, and 30 μM final DHA concentrations that were tested, we selected the lowest one which could cause inhibition of proliferation as well as differentiation for conducting all the experiments. A dose of 50 µM DHA was lethal for EoL-1 cells which were lysed after 3 days of exposure and this concentration was not included in the study. Sodium butyrate (B 5887, SIGMA-ALDRICH, Darmstadt, Germany), a well-known differentiating agent for EoL-1 cells, was used as a positive control in the study.

### 2.3. Cell Viability Measurement by MTT Assay

For the MTT assay, 50 μL of cell suspension from each tube was aliquoted into duplicate wells of 96-well plates for each concentration point. Immediately, 20 μL of the MTT reagent (Cell Titer 96 AQ_ueous_One Solution Reagent, Promega Madison, Fitchburg, WI, USA) was added in each well in the dark. The plate (covered in order to avoid exposure to light) was then transferred back to the incubator at 37 °C/5% CO_2_ for 60 min and then directly to an Elisa reader (MWG Biotech AG Ebersberg, Germany). The photometric measurement was performed at 490 and 630 nm, according to the manufacturer′s instructions.

### 2.4. Eosin/Hematoxylin EoL-1 Cell Staining

EoL-1 cells were treated with 10 μM DHA or 500 μM butyrate. Untreated cells were used as control. Cells were collected by centrifugation at 500× *g* for 10 min. Then, the pellet was spread properly on the surfaces of two glass slides. After one minute, the next steps involved sequential dipping in 96% ethanol solution for 15 min and washed in water 3–4 times; hematoxylin (Hematoxylin solution, Merck, KGaA, Darmstadt, Germany) for 10 min and washed in water 3–4 times; a bath with 96% ethanol acidified with 1% HCl 2–3 times; eosin (Eosin Y 1% alcoholic solution, Biostain, Molekula Atom Scientific LTD, Cheshire, United Kingdom) for 3 min and washed in water 3–4 times; washed in 70% ethanol 6–7 times; 80% ethanol 6–7 times; acetone 2–3 times; xylene (xylene, Klinipath, Duiven, Netherlands) for 5 min. The slides were then transferred directly to the microscope for observation.

### 2.5. Total RNA Isolation from EoL-1 Cells and qRT-PCR Analysis

For qRT-PCR experiments, cell pellet was lysed after the removal of the supernatant, with the addition of lysis buffer solution provided by the NucleoSpin RNA II kit (Macherey-Nagel, GmbH & Co. KG, Dueren, Germany). Total RNA was isolated according to the manufacturer’s instructions. RNA integrity and purity was checked electrophoretically and verified with the criterion of an OD_260_/OD_280_ absorption ratio >1.7.

qRT-PCR was performed using KAPA SYBR^®^ FAST One-Step qRT-PCR Kit (Wilmington, MA, USA), using forward and reverse primers from QIAGEN (Redwood City, CA, USA) for *PTAFR*, *MYC*, *PLA2G4A*, and *GAPDH* human genes, with the last used as the reference gene.

Total RNA (100 ng) in a 20 μL total volume was first incubated at 42 °C for 10 min to synthesize cDNA, heated at 95 °C for 4 min to inactivate the reverse transcriptase, and then subjected to 35 thermal cycles (95 °C for 2 s, 60 °C for 20 s) of PCR amplification and 35 cycles from 65 °C to 95 °C (0.5 °C increment) for melting curve analysis using an MJ Mini Opticon (Bio-Rad, Hercules, CA, USA).

The size of the amplification products from each different set of primers was confirmed by agarose gel electrophoresis in order to verify the length of the product before calculating the qRT-PCR data with Biorad CFX Manager software. Relative quantitation of qPCR data was carried out according to the method of Pfaffl [22]. qRT-PCR results were calculated as fold-increase of target gene mRNA versus fold-increase in *GAPDH* mRNA.

### 2.6. Cell Cycle Arrest by Flow Cytometry

The EoL-1 cells pellet (5 × 10^6^ cells) of each condition (in the presence or absence of DHA concentrations) was resuspended into 1ml PBS and centrifuged at 500× *g* for 10 min. Finally, the cells were resuspended into 300 μL PBS and 700 μL of absolute cold ethanol (96%) was added. The samples were stored at −20 °C.

Prior analysis of the cell cycle, cells were collected and washed three times with PBS. Subsequently, the cells were resuspended in 1 mL PBS.

Then, 10 μL RNaseA (20mg/mL, SIGMA) was added followed by incubation at 37 °C for 30 min. 4 μL PI (propidium iodide 1mg/mL, SIGMA) stain was added and the samples were left on ice (4 °C) for 30 min in the dark. Finally, analysis of the phases of the cells cycle by flow cytometer (BD Biosciences, Franklin Lakes, NJ, USA) was performed. Single cells were detected in flow cytometry application by gating for events based on forward scatter and side scatter populations, and by using the properties of DNA staining by propidium iodide. In each run, a total cell count of 30,000 cells was set.

### 2.7. Cytokines Levels by ELISA

The production of IL-6 and TNF-α of EoL-1 cells after DHA or butyrate treatment was measured by enzyme-linked immunosorbent assay (ELISA) in the cells’ medium, after centrifugation at 500× *g* and discarding the cell pellet, with a kit according to the manufacturer’s protocol (R&D Systems).

### 2.8. Total PLA_2_ and PAFAH Activities in EoL-1

Total Ca^2+^-dependent PLA_2_ activity in cell homogenates and cell supernatants was measured by a fluorimetric assay developed in our laboratory [23,24]. The incubation mixture contained 240 μL of 10 mM Tris-HCl buffer solution, 2 mM Ca^2+^, pH 7.4, and 5 μM C_12_-NBD-PC (1-palmitoyl-2-(12-[amino]dodecanoyl)-sn-glycero-3-phosphocholine, AVANTI) as the fluorescent phospholipid substrate. The reaction started with the addition of the source of the enzyme containing 5–15 μg of total protein. Incubation took place for 4 hrs. Excitation and emission wavelengths were adjusted to 475 and 535 nm, respectively. The enzymatic activities were calculated in real time from the slope of the response curve by using C_12_-NBD-FA as internal standard. For PLA_2_ and PAFAH activities, the sensitivity of the assay used is 0.1 pmole C_12_- or C_6_-NBD-FA/h/μg protein.

The previous experimental protocol was also applied for the determination of PAFAH activity with the only difference that the substrate was an NBD-fluorescently labeled phosphatidylcholine (C_6_-NBD-PC) with a short fatty acyl chain (C6) at the *sn*-2 position. In the absence of Ca^2+^(or presence of 10 mM EDTA), C_6_-NBD-PC is known to be hydrolyzed by phospholipase A_2_ with preference for short acyl chains such as PAFAH. In this respect, applying the above methodology we could discriminate between Ca^2+^dependent PLA_2_ activities with preference for long acyl chains and Ca^2+^independent PLA_2_ activity with preference for short acyl chains such as PAFAH [23].

## 3. Results and Discussion

### 3.1. Effect of DHA on the Proliferation of EoL-1 Cells

In order to clarify whether DHA inhibited the proliferation of EoL-1 cells, the number of DHA-treated cells in certain time points and various DHA concentrations was counted by trypan blue staining. EoL-1 cells proliferated in a time-dependent manner during 8 days of incubation in the absence of any inhibitory agent.

The number of EoL-1 cells was doubled after 3 days of incubation (logarithmic phase of growth) in the complete medium at 37 °C/5% CO_2_. As shown in Figure 1A, in the presence of DHA, an inhibitory effect on the proliferation of EoL-1 cells was observed in a significant dose-dependent manner. Significant inhibition upon the proliferation of EoL-1 cells was observed on day 3 and 4 (Figure 1A). n-Butyrate was used as a positive control because it is known that it comprises a differentiating agent which inhibits proliferation of EoL-1 cells. Five hundred μM n-butyrate also significantly inhibited the proliferation of EoL-1 cells (Figure 1A).

Dead cell counting by Trypan blue showed no significant alteration in the ratio of EoL-1 viable to dead cells from day 1 to day 4 of 10 μM DHA treated EoL-1 cells. A similar effect was also exhibited by butyrate. These results suggest that no direct toxic effect was exerted by 10 μM DHA but reflected reduced proliferation (Figure 1B). On the contrary, for 20 and 50 μM DHA concentrations, the ratio of viable to dead cells was decreased significantly on day 3 (Figure 1B). The observation indicates that DHA concentrations over 20 μM were toxic to EoL-1 cells.

Similar results were obtained by employing the MTT assay in order to study the effects of DHA on EoL-1 cell line viability. As shown in Figure 1C, for treatment with 5 μM DHA the loss in active metabolism was not significant (97%) as compared to control cells. However, cells treated with 10 μM DHA showed a loss of active metabolism (86%), and the effect was dose-dependent (78% for 20 μM and 51% for 50 μM DHA). The LD_50_ after 3 days of incubation of EoL-1 cells was found at 45 μM DHA.

The combined effect of DHA and butyrate was also studied at high concentrations. The inhibitory effect of DHA on the proliferation of EoL-1 cells was higher in EoL-1 cells preincubated with butyrate. It was observed that 50 μM DHA caused a similar decrease in EoL-1 cells’ viability (51.6% ± 6.2) after 3 days of incubation as 500 μM butyrate (43.7% ± 3.2). When DHA and butyrate were added simultaneously in EoL-1 cell culture, the 3-day viability reached only 17.8% (Figure 1D). The combination index (CI value) for those two substances was <1 (calculated at 0.485 ± 0.045 by CompuSyn software), thus indicating a synergistic effect [25].

Thus, up to 10 μM concentration, DHA did not affect the viability of EoL-1 cells. Above 10 μM, both DHA and butyrate serving as differentiating agent positive control, inhibited the proliferation of EoL-1 cells in a significant time and dose-dependent manner. The combined effects of the two substances (DHA + Butyrate) were almost equal to the sum of the effects of each individual substance indicating an additive effect on EoL-1 cells viability. The antineoplastic activity of DHA has been indicated in many studies. DHA induces apoptosis and differentiation in human melanoma cells in vitro [26], helps prevent colorectal cancer [19], induces apoptotic cell death in human cancer cells either alone or in combination with conventional therapies, and causes selective cytotoxicity towards cancer cells with little or no toxicity on normal cells [27]. DHA can induce apoptosis in tumor cells in vitro and in vivo, in a dose- and time-dependent manner in tumor cell lines derived from a wide range of solid tumors. Apoptosis induced by DHA has been also described in cancer cell lines derived from hematological tumors such as myeloid and lymphoid leukemias and lymphomas, as well as multiple myeloma (for review see [26]).

### 3.2. Effect of DHA on the Expression of MYC-mRNA in EoL-1 Cells

The expression of the *MYC* protooncogene in EoL-1 cells was examined by qRT-PCR in order to investigate whether the differentiating activity was associated with a reduction of *MYC* mRNA levels. DHA caused a reduction on *MYC* mRNA, which was time dependent. *MYC* mRNA levels were gradually decreased from 30 min to day 1, day 2, and day 3 of DHA incubation with reductions of 20%, 71%, 79%, and 83%, respectively (Figure 2).

DHA inhibited proliferation and induced differentiation of EoL-1 cells through the suppression of *MYC* gene expression. The protein encoded by the *MYC* gene is a multifunctional, nuclear phosphoprotein that plays a role in cell cycle progression, apoptosis, and cellular transformation. *MYC* is a transcription factor which alters the expression of hundreds of target genes and has long been known to be among the most frequently deregulated oncogenes in human cancer [28]. Mutations, overexpression, rearrangement and translocation of this gene have been associated with a variety of hematopoietic tumors, leukemias and lymphomas, while high expression levels of *MYC* preserve the undifferentiated phenotype of the cells [29]. Additionally, the differentiation of EoL-1 cells by cytokines is associated with downregulation of *MYC* gene expression [11].

### 3.3. Effect of DHA on the Morphology of EoL-1 Cells

Most EoL-1 cells are small and round with a round nucleus and no cytoplasmic granules. They have high nuclear/cytoplasmic ratio and lack nuclear lobulation [30]. DHA-treated EoL-1 cells showed, apart from reduced proliferation rates (Figure 3A,B), an asymmetrical round shape, nuclear shrinkage, and formation of crystalloid cores (or inclusions) (Figure 3D) which are characteristic of mature peripheral blood eosinophils. On the other hand, butyrate-treated EoL-1 cells grew in clusters and exhibited an asymmetrical round shape, nuclear shrinkage, and lobulation (Figure 3E).

### 3.4. DHA Induced Cell Cycle Arrest in G0/1 Phase of EoL-1

In order to investigate whether the decreased cell number of EoL-1 cells observed with low DHA concentrations (10 μM) were due to cell cycle growth arrest, flow cytometric analysis on the 3rd day of DHA treatment was performed. The EoL-1 pellets were stained with propidium iodide before the cytometric analysis. Cell cycle analysis showed a shift of cells into the G0/G1 phase from the S/G2 phase following treatment with 10 μM DHA (G0/G1 phase: 71.81% ± 6.38% for DHA vs. 63.57% ± 4.79% for control (Figure 4A,D), G2 phase: 9.85% ± 4.01%for DHA vs. 16.32% ± 6.10% for control (Figure 4B,E), and S + G2 phases: 19.94% ± 8.56% for DHA vs. 27.30% ± 13.37% for control (Figure 4C,E). These results confirmed that the decreased proliferation of DHA-treated EoL-1 cells were due to G0/1 growth arrest.

### 3.5. DHA did not Affect the Expression of FIP1L1-PDGFRα Gene in EoL-1 Cells

In order to test the hypothesis in which DHA inhibits the growth of leukemic eosinophils through targeting of the disease-related oncoprotein *FIP1L1-PDGFRα*, qRT-PCR experiments were conducted. The *FIP1L1-PDGFRα* mRNA expression in EoL-1 cells was quantified versus *ABL* mRNA as a reference. However, no alteration in *FIP1L1-PDGFRα* mRNA expression levels was observed in 10 μM of DHA treated EoL-1 cells indicating that growth arrest was not through downregulation of *FIP1L1-PDGFRα* gene expression.

### 3.6. Effect of DHA on the Expression of PTAFR-mRNA in EoL-1 Cells

Our previous experiments confirmed that DHA had antiproliferative effects on EoL-1 cells similar to butyrate, which is known to induce EoL-1 cells differentiation accompanied by overexpression of human platelet-activating factor receptor gene (*PTAFR*). *PTAFR* is a specific receptor for PAF, located on eosinophils plasma membrane. The biological effects of PAF are mediated by binding and thus activation of *PTAFR*. However, *PTAFR* expression is modulated by many agents, including those which induce cell differentiation, and thus is considered as a marker of differentiation for EoL-1 cells [14]. Additional experiments were designed to investigate whether DHA antiproliferative effects on EoL-1 cells were associated with gene transcription of *PTAFR*.

qRT-PCR data revealed that EoL-1 control cells expressed a relatively low level of *PTAFR*-mRNA. The amount of *PTAFR-*mRNA was increased by 1.2 ± 0.20 and 1.59 ± 0.11 fold versus control, after 2 and 3 days of exposure to DHA, respectively. No significant alteration was observed after 24 h of DHA treatment (Figure 5A). *PTAFR* mRNA expression at 10, 20, and 30 μM of DHA concentrations revealed that the phenomenon was dose dependent (Figure 5B). Moreover, butyrate (used as a positive control in our study) increased the expression of *PTAFR* gene by 10.53 ± 0.51 fold, versus control (Figure 5C).

For the first time, the present study provided evidence that DHA-treated EoL-1 cells were induced to differentiate towards the eosinophilic phenotype, during which the expression of PAF-receptor gene was increased. The increased expression of *PTAFR* is a marker of EoL-1 differentiation [14]. Differentiation of EoL-1 cell line is also induced by the culture supernatant of a human ATL cell line (HIL-3 sup) which induces EoL-1 cells to respond to PAF [31]. In addition, overexpression of *PTAFR* is known to augment inflammatory responses [32].

PAF is a potent phospholipid-derived inflammatory mediator [33] produced and secreted by several types of cells, including mast cells, monocytes, tissue macrophages, platelets, eosinophils, endothelial cells, and neutrophils. PAF is implicated in platelet aggregation and activation of the inflammatory response in which PAF signaling has an established role [34,35]. Regarding eosinophils, the generation and release of PAF is a consequence of normal eosinophil chemotactic activation in order to function in inflammatory and allergic reactions in which eosinophils participate [36]. Normal eosinophils are recruited from the blood to sites of inflammation under the influence of several eosinophil chemotactic factors, such as eosinophil chemotactic factor of anaphylaxis, leukotriene B4, and PAF. Eosinophils themselves also secrete PAF, which leads to further accumulation of eosinophils [10]. The biological effects of PAF are mediated mainly by binding and activating *PTAFR* [37]. It is known that *PTAFR* gene encodes a seven-transmembrane G-protein-coupled receptor for platelet-activating factor that localizes to lipid rafts and/or caveolae in the cell membrane [38].

Various mechanisms have been proposed to underlie the effect by which DHA (or n-3 PUFAs) induce differentiation oftumor cells. These are: (1) DHA incorporation into various cellular lipids, affecting significantly membrane structure and fluidity, function of membrane receptors and lipid-mediated signaling [2]. (2) DHA and its intracellular oxidative metabolites bind to nuclear receptors including PPARγ, leading to altered gene expression [39] (3) DHA generates lethal levels of intracellular oxidative metabolites leading to oxidative stress [26].

### 3.7. Enzymatic Activities of PAFAH in DHA-Treated EoL-1 Cells

The enzyme activity of PAFAH was measured in DHA-treated EoL-1 cells in order to investigate whether DHA had an effect on its activity since PAFAH hydrolyses and deactivates PAF [40], thus interrupting the PAF/*PTAFR* signaling pathway. We applied the fluorescent methodology in the presence of 10 mM EDTA and C_6_-NBD-PC as substrate which is known to be hydrolyzed by phospholipase A_2_ with preference for short acyl chains such as PAFAH.

PAFAH activity was detected in all samples of DHA-treated EoL-1 cells homogenates and culture supernatants. Butyrate-treated EoL-1 cells showed decreased levels of PAFAH activity (1.73 ± 0.77 pmole C6-NBD-FA/h/μg protein) as compared to untreated control EoL-1 cells (5.77 ± 2.44 pmole C6-NBD-FA/h/μg protein). Ten μM DHA-treated EoL-1 cells showed no significant alteration on the levels of PAFAH activity as compared to control EoL-1 cells. However, EoL-1 cells preincubated with 20 μM DHA had lower PAFAH activity (3.12 ± 1.82 pmole C6-NBD-FA/h/μg protein) than control. Thirty μM DHA had no effect on PAFAH activity of EoL-1 cells homogenate. In EoL-1 cell culture supernatants, the profile of PAFAH activity levels of DHA- or butyrate-treated versus untreated cells were similar to that of EoL-1 cells homogenates (Figure 6A).

PAFAH activity was detected in low levels in the samples of 20 and 30 μM DHA-treated EoL-1 cell homogenates as well as in culture supernatants, with a similar profile. Butyrate-treated EoL-1 cells showed decreased levels of PAFAH activity as compared to untreated control EoL-1 cells. However, the role of this enzyme is still not clear [32]. Studies in cell lines and in mouse models show the diversity of functions of PAFAH in cancer, and the potential of PAFAH transcripts, protein, and/or activity levels to become cancer biomarkers and therapeutic targets [41].

### 3.8. Enzymatic Activities of PLA_2_ in DHA-Treated EoL-1 Cells

Phospholipase A_2_ enzyme family members contribute to the biosynthesis of PAF. Biosynthesis of PAF occurs via two different routes: the de novo and the remodeling pathways. The remodeling pathway of PAF biosynthesis is the major pathway during inflammation [42] and occurs in two steps. First, members of the phospholipase A_2_ family generate an intermediate, lyso-PAF, from membrane phospholipids. Lyso-PAF is then converted to PAF by the action of lyso-PAF-acetyltransferase. PAF is rapidly hydrolyzed and degraded to an inactive metabolite, lyso-PAF, by the enzyme PAF acetylhydrolase, the activity of which has shown to correlate inversely with PAF levels. Many isoforms of PLA_2_ enzymes contribute to PAF synthesis via the remodeling pathway. These are Ca^2+^-dependent isoforms such as cytosolic phospholipase A_2_ (cPLA_2_) and secreted phospholipase A_2_ (sPLA_2_) as well as Ca^2+^-independent iPLA_2_, with cPLA_2_ (also known as Group IV PLA_2_, gene *PLA2G4A*) having a major role on PAF biosynthesis [43].

However, all of the previously mentioned PLA_2_ isoenzymes have a common characteristic, the preference to hydrolyze long fatty acyl chain esterified at the *sn*-2 position of the phospholipids. Total EoL-1 phospholipase A_2_ activity specific to hydrolyze long acyl moieties was measured by a fluorimetric method using a phosphatidylcholine with a long (C12) fluorescent lipid moiety esterified at the *sn*-2 position as substrate. The method is suitable for the simultaneous measurement of total cPLA_2_, sPLA_2_, as well as iPLA_2_ activities in the presence or absence of Ca^2+^in the reaction mixture [23].

In the presence of 10 μM DHA, EoL-1 intracellular phospholipase A_2_ activity was increased (3.41 ± 0.12 pmole C_12_-NBD-FA/h/μg protein) as compared to untreated EoL-1 control cells (0.44 ± 0.44 pmole C_12_-NBD-FA/h/μg protein). Subsequently, total PLA_2_ activity levels were reduced after incubation with 20 and finally to 30 μM of DHA (1.21 ± 0.49 pmole C_12_-NBD-FA/h/μg protein for 20 μM DHA and 0.87 ± 0.70 pmole C_12_-NBD-FA/h/μg protein for 30 μM DHA). Butyrate-treated EoL-1 cells did not show any PLA_2_ activity. In DHA-treated EoL-1 cell culture supernatants, PLA_2_ activity was not detected. In EoL-1 control cells, supernatant PLA_2_ activity of 2.04 ± 2.04 pmole C_12_-NBD-FA/h/μg protein was measured (Figure 6B).

It was clearly shown that total Ca^2+^-dependent PLA_2_ activity was only detected in DHA-treated and not in butyrate-treated EoL-1 cells homogenates. This could possibly be attributed to different routes of differentiation which butyrate and DHA might induce. The levels of total intracellular PLA_2_ activity were higher in DHA-treated versus untreated EoL-1 cells. Thus, DHA at low concentration (10 μM) caused a large increase in total Ca^2+^-dependent PLA_2_ activity of EoL-1 cells, suggesting that more than one PLA_2_ isoform could exist. At higher DHA concentrations (20 and 30 μM) the total PLA_2_ activity levels remained low.

In culture medium, PLA_2_ activity was only detected in untreated EoL-1 cells. This finding reflects the secretory types of PLA_2_ (sPLA_2_), which are known to play an important role in inflammation as well as tumorigenesis [44,45,46]. The plasma level of sPLA_2_ group IIA is increased in patients with different types of malignancies, but it is unclear whether the upregulated enzyme expression is directly related to tumorigenesis or this is a consequence of tumor-associated inflammation [47]. The role of sPLA_2_s in cancer has been associated with their enzyme activity and ability to participate in the release of potent biologically active lipid mediators, in particular arachidonic acid-derived eicosanoids. Phospholipase A_2_ mediates the rate-limiting step in the formation of eicosanoids which promote tumorigenesis by stimulating cell proliferation and cell survival, by abrogating apoptosis and by increasing local inflammation and angiogenesis [48]. It is known that secreted isoforms of PLA_2_ (sPLA_2_ group IIA and sPLA_2_ group X) are expressed in human eosinophils and located in specific granules or phagosomes [49,50,51].

### 3.9. Effect of DHA on the Expression of IL-6 and TNF-a in EoL-1 Cell Culture Medium

The inflammatory cytokines IL-6 and TNF-α were measured in the culture medium of DHA-treated EoL-1 cells in order to investigate the DHA effect. After 3 days of incubation, the levels of IL-6 in the medium of 10 μM DHA-treated EoL-1 cells were found increased at 3.14 ± 0.18 pg IL-6/mL as compared to control cells (2.19 ± 0.28 pg IL-6/mL). With 20 μM DHA, the levels of IL-6 (2.33 ± 0.22 pg IL-6/mL), did not differ significantly from the control. Finally, DHA (30 μM) and butyrate (500 μM), caused significantly reduced levels (1.15 ± 0.22 and 1.31 ± 0.05 pg IL-6/mL, respectively), of IL-6 in the cell supernatants, compared to the control (Figure 7A). TNF-αwas significantly increased in the medium of all DHA- and butyrate-treated cells (3.27 ± 0.74, 2.61 ± 0.67, and 2.92 ± 0.43 after treatment with 10, 20, 30 μM DHA and 3.31 ± 0.13 after butyrate incubation, respectively) compared to the control (Figure 7B).

Omega-3 fatty acids (e.g., eicosapentaenoic acid and docosahexaenoic acid) have anti-inflammatory properties and are known to reduce the production of inflammatory enzymes and cytokines, including COX-2, tumor necrosis factor (TNF)-α, and interleukins (IL-6, IL-1β), in inflammatory conditions [52,53]. However, in the present study, where the differentiation potential of DHA towards EoL-1 cells was investigated, we found that together with the eosinophilic type induction, the inflammatory signaling pathway of PAF/*PTAFR* could become more functional [54]. The increased levels of the inflammatory cytokines IL-6 and TNF-α after three days of cell treatment with 10 μM of DHA support this concept.

At higher doses of DHA (20 and 30 μM), only TNF-αremained increased compared to control, while IL-6 reached similar or lower levels. It could be suggested that 10 μM DHA could enhance the inflammatory function of EoL-1 cells during differentiation and this can be observed through increased IL-6 and TNF-α production. The lower levels of IL-6 produced at higher doses of DHA (20 and 30 μM) could be due to extremely low cell viability in such concentrations (Figure 1A–C)

Butyrate, as a differentiating agent of EoL-1 cells, caused decreased IL-6 production and increased TNF-α production. The increased production of TNF-α agrees with the hypothesis that inflammatory cytokines are produced during EoL-1 cell differentiation [55,56].

Based on our experimental data, it seems that DHA exhibits a differentiating activity towards EoL-1 cancer cells, turning them to a phenotype of non-malignant eosinophils that produce low levels of IL-6 and TNF-α pro-inflammatory cytokines, compatibly to their function as immunological cells.

### 3.10. Effect of DHA on the Expression of PLA2G4A mRNA in EoL-1 Cells

In order to test whether the increased activity levels of PLA_2_ recorded in DHA treated EoL-1 cells was due to increased expression, the mRNA levels of *PLA2G4A* were measured. The *PLA2G4A* gene encodes the cPLA_2_ isoform (also known as Group IV PLA_2_) which has a major role on PAF biosynthesis [42].

An increase in the level of *PLA2G4A* mRNA was observed after incubation of EoL-1 cells with 10 and 20 μM DHA (1.27 ± 0.10 and 1.50 ± 0.11 fold versus untreated, respectively). However, with 30 μM DHA, *PLA2G4A* mRNA remained at the same level as those with 20 μM DHA (1.51 ± 0.15 fold, Figure 5B). On the contrary, butyrate caused a reduction in the expression level of *PLA2G4A* mRNA versus control cells (Figure 4C).

DHA induced both *PTAFR* and *PLA2G4A* gene expression in EoL-1 cells. *PLA2G4A* mRNA was increased by approximately 50% after treatment with 20 μM DHA and remained at high levels up to 30 μM DHA. Butyrate, apart from total PLA_2_ activity reduction, also decreased *PLA2G4A* mRNA expression levels. Those differences on the effects of DHA and butyrate regarding *PLA2G4A* could be attributed to the fact that DHA is a polyunsaturated fatty acid with a very long chain that could be metabolized intracellularly to several oxidized molecules [26], whereas butyrate is a saturated fatty acid with very short chain length and cannot be further oxidized to yield such products.

In the present work, we demonstrated for the first time the antiproliferative and differentiating effects of docosahexaenoic acid (an omega-3 highly polyunsaturated fatty acid) on the EoL-1 leukemic cell line. We presented that EoL-1 cell differentiation was related to the downregulation of *MYC* protooncogene mRNA and associated with the induction of platelet activating factor receptor (*PTAFR*) gene expression, the increase in cytosolic phospholipase A_2_ activity and gene expression, and the decrease of secreted PLA_2_ as well as PAFAH activities. All these findings indicate that the induction of *PTAFR* expression in DHA-treated EoL-1 cells could be linked to concurrent activation of intracellular signaling pathways that result in the expression and activation of cytosolic PLA_2_ (cPLA_2_) possibly towards PAF production through remodeling pathway. In this way, it is possible that the PAF/*PTAFR* pathway, which is active in normal eosinophils, is functional with the increased expression of *PTAFR* by increasing cPLA_2_ gene expression and activity parallel to the decrease in PAFAH activity. Future research directions may also be the investigation of DHA uptake from EoL-1 cells, trafficking, distribution, or/and metabolism of the lipid moiety, parallel to intervention experiments with pharmacological inhibitors or siRNA which could be conducted towards the direction of elucidating the biochemical pathways or mechanisms which participate in DHA inhibition of proliferation of EoL-1 cells.

## 4. Conclusions

The main conclusions of the present work were that DHA had antiproliferative and differentiating properties on EoL-1 leukemic cells. The DHA effect was related to cellular events such as downregulation of the *MYC* oncogene and overexpression of the *PTAFR* and *PLA2G4A* genes as well as altered enzymatic activities of PLA_2_ and PAFAH. The inflammatory signaling pathway of PAF/*PTAFR*, which is characteristically functional in normal eosinophils, could become more functional in DHA-treated EoL-1 cells. DHA is a natural substance which could be tested as a future therapeutic tool against F/P + hypereosinophilic syndrome.

## Figures and Tables

**Figure 1 nutrients-11-00574-f001:**
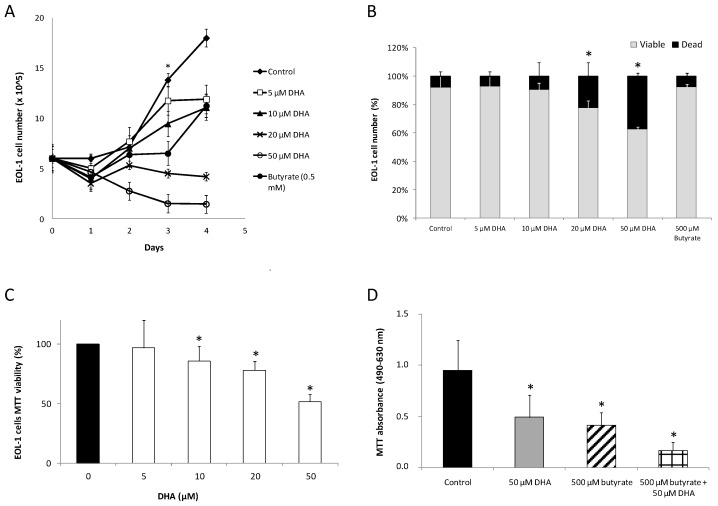
(**A**) Effect of various DHA (docosahexaenoic acid) concentrations on the growth of EoL-1 cells. EoL-1 cells were incubated for the periods indicated at 37 °C/5% CO_2_ in the absence or presence of various concentrations of DHA (5, 10, 20, 50 μM) with 500 μM butyrate as a positive control. The cells were then collected, stained with trypan blue, and counted. * Statistical significance of *p* < 0.01 vs. the corresponding control in all DHA concentrations on the third day of growth (*n* = 5, in duplicates, *t*-test analysis for independent samples, Statistica Software). (**B**) Viable to dead cell counting with trypan blue staining. The starting EoL-1 cell culture consisted of 1.5 × 10^6^ cells/mL complete medium, and were left to grow for 3 days at 37 °C/5% CO_2_ in the absence of DHA (control cells) or presence of 5, 10, 20, 50 μM DHA or 500 μM butyrate. Cell counting was performed after 72 h. Values represent the average of the percentage of viable or dead cells number in the medium ± SD. * Statistically significant difference of *p* < 0.01 versus control (*n* = 3 in duplicates, *t*-test analysis for independent samples, Statistica Software). (**C**) EoL-1 cells viability. Effect of various DHA concentrations (5, 10, 20, 50 μM) on the viability of EoL-1 cells, measured by MTT assay. The viability of the ΕOL-1 cells was reduced to 97, 86, 78, and 51% when incubated with 5, 10, 20, and 50 μM DHA, respectively. Values represent the means ± SD. * Statistical significant difference of *p* < 0.01 versus control (*n* = 3, in duplicates, *t*-test analysis for independent samples, Statistica Software). (**D**) EoL-1 cells MTT viability in combination with butyrate. Effect of 50 μM DHA or 500 μM butyrate or combination of both on the viability of EoL-1 cells, measured by MTT assay. Values represent the means ± SD. * Statistical significant difference of *p* < 0.01 versus control (*n* = 3, in duplicates, *t*-test analysis for independent samples, Statistica Software).

**Figure 2 nutrients-11-00574-f002:**
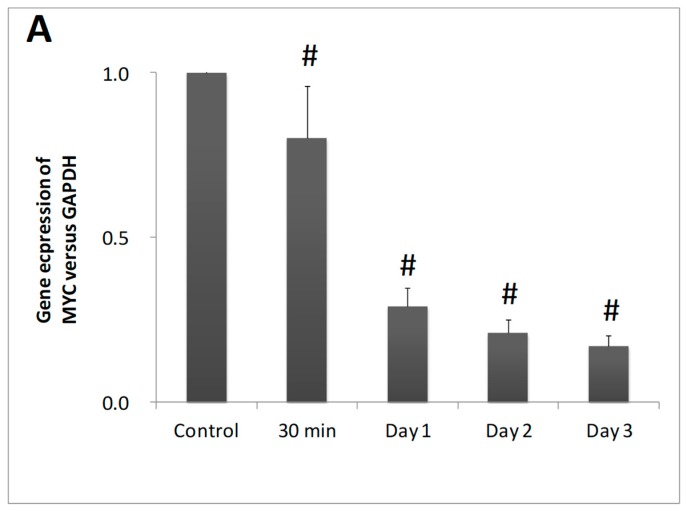
Effect of DHA treated EoL-1 cells upon *MYC*-mRNA levels. *MYC*-mRNA levels from EoL-1 cells in the absence (control) or presence of 10 μM DHA by qRT-PCR analysis (*n* = 3, in duplicates). The symbol # denotes statistically significant differences of *p* < 0.05 versus control. *t*-Test analysis for independent samples was applied by Statistica software.

**Figure 3 nutrients-11-00574-f003:**
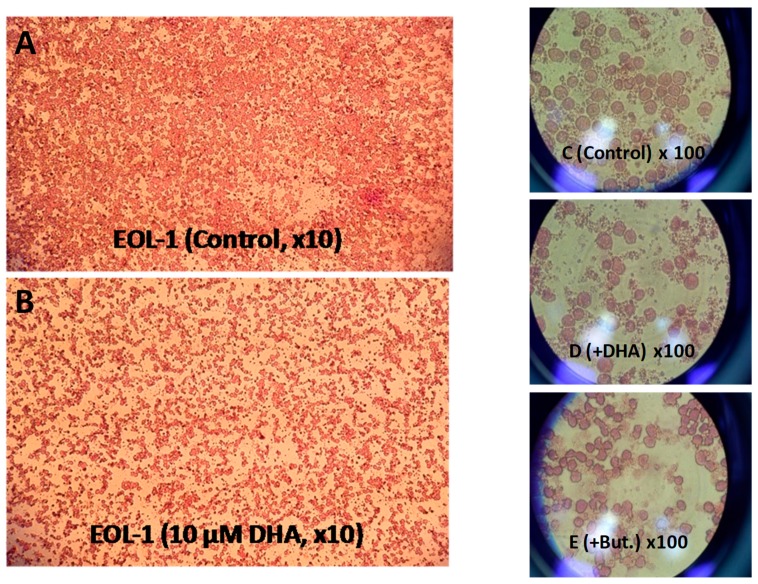
Light microscopy of EoL-1 cell culture. (**A**) in the absence of DHA (Control EoL-1 cell culture with ethanol in the medium) and (**B**) in the presence of 10 μM DHA, after 3 days of incubation at 37 °C/5% CO_2_. (**C**) Effect of DHA on morphology of EoL-1 cells: control EoL-1 cells × 100 magnification. (**D**) 10 μM DHA treated EoL-1 cells × 100 magnification. (**E**) 500 μM Butyrate treated EoL-1 cells × 100 magnification.

**Figure 4 nutrients-11-00574-f004:**
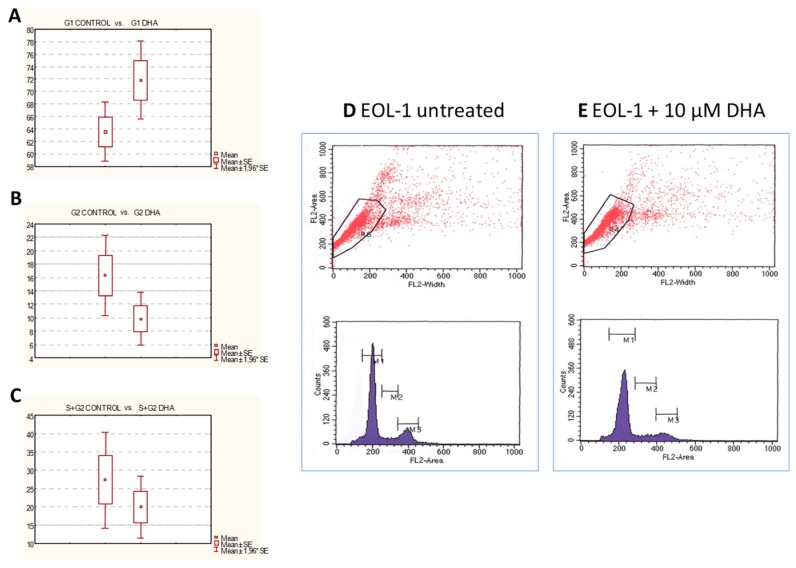
Cell cycle analysis of EoL-1 cells by flow cytometry with propidium iodide staining. Cell cycle analysis showed a shift of cells into the G0/G1 phase from the S/G2 phase following treatment with 10 μM DHA. (**A**) G0/G1 phase: 71.81% ± 6.38% for DHA vs. 63.57% ± 4.79% for control (*p* < 0.01). (**B**) G2 phase: 9.85% ± 4.01% for DHA vs. 16.32% ± 6.10% for control (*p* < 0.01). (**C**) S + G2 phases: 19.94% ± 8.56% for DHA vs. 27.30% ± 13.37% for control (*p* < 0.05). DNA histograms of cell cycle distribution: (**D**) Untreated EoL-1 cells. (**E**) DHA (10 μM for 72 h) treated EOL-1 cells. Representative graph.

**Figure 5 nutrients-11-00574-f005:**
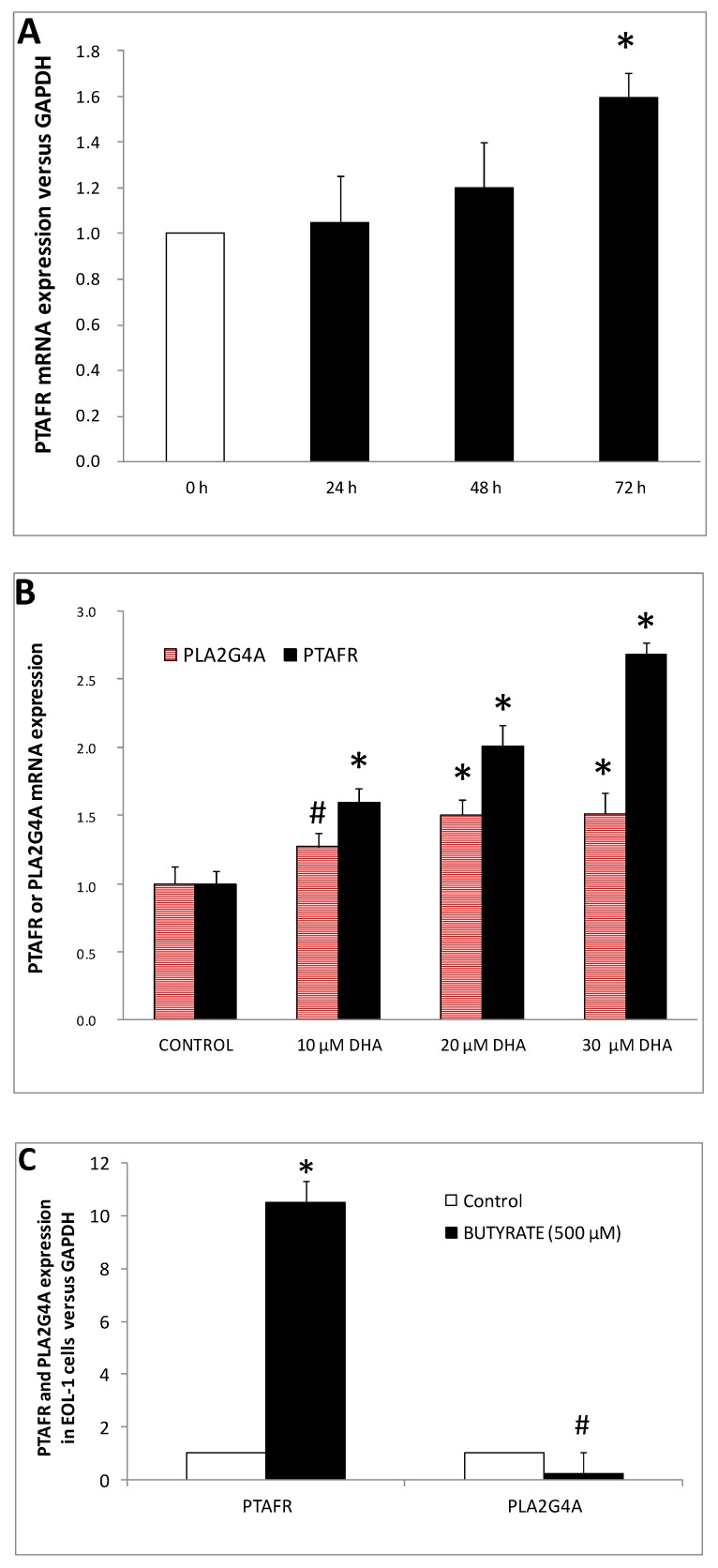
DHA induced *PTAFR* and *PLA2G4A* gene expression on EoL-1 cells. (**A**) Time course in DHA (10 μM) induced *PTAFR* mRNA expression (*n* = 4, in triplicates). (**B**) The levels of *PTAFR* and *PLA2G4A* mRNA expression were measured after 3 days (*n* = 3, in duplicates) of treatment with 10, 20 and 30 μM DHA. (**C**) Butyrate induced expression of *PTAFR* and *PLA2G4A*-mRNA (*n* = 3, in duplicates). The symbols # and * denote statistically significant difference of *p* < 0.05 and *p* < 0.01, respectively as compared to control. *t*-Test analysis for independent samples was applied by Statistica software.

**Figure 6 nutrients-11-00574-f006:**
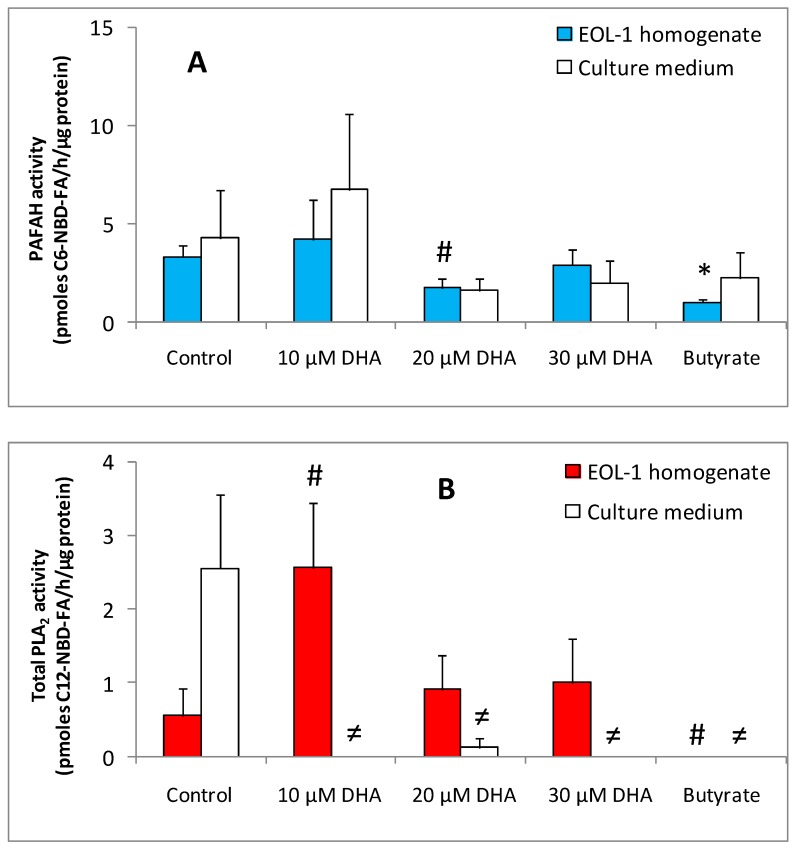
Effect of DHA treatment upon PAFAH (PAF-Acetylhydrolase) and PLA_2_ enzymatic activities of EoL-1 cells. (**A**) PAFAH activities of EoL-1 homogenates and supernatant of 3 days treated (10 μM DHA or 500 μM butyrate) and untreated cells (*n* = 3, in duplicates). (**B**) PLA_2_ activities of EoL-1 homogenates and supernatant of 3 days treated (10 μM DHA or 500 μM butyrate) and untreated cells (*n* = 3, in duplicates). PLA_2_ activity was not detectable in culture medium at 10, 30 μM DHA and in butyrate. The symbols # and ≠ denotes statistically significant differences of *p* < 0.05 for homogenates and culture medium, respectively, versus control. * denote statistical significant difference of *p* < 0.01 as compared to control. *t*-Test analysis for independent samples was applied by Statistica software.

**Figure 7 nutrients-11-00574-f007:**
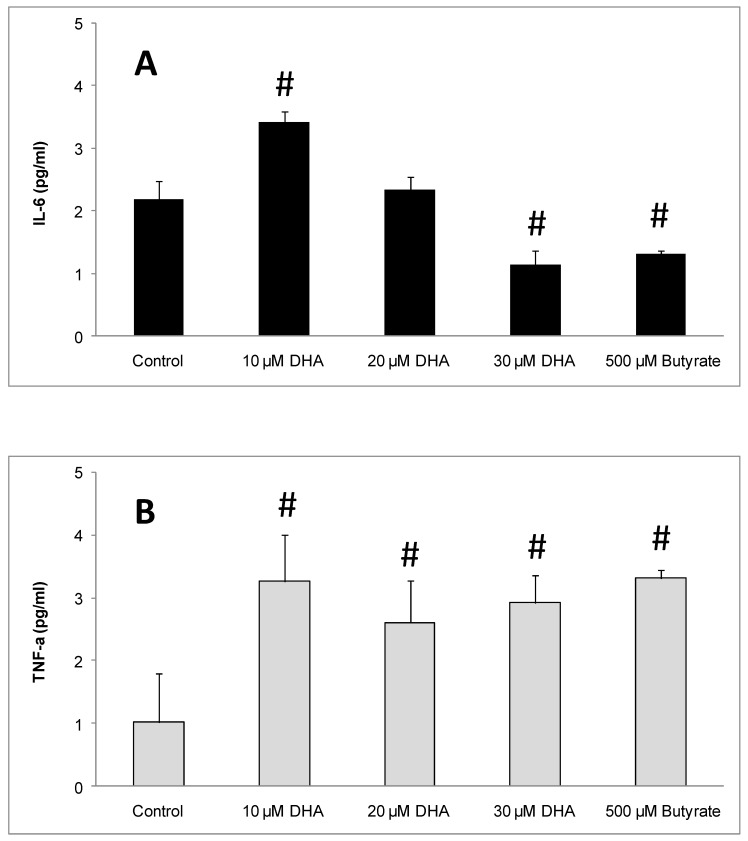
Effect of DHA on EoL-1 cells and IL-6 and TNF-α production. (**A**) IL-6 levels from EoL-1 cells culture medium in the absence (control) or presence of 10, 20, 30 μM DHA and 500 μM butyrate by ELISA analysis (*n* = 3, in triplicates). (**B**) TNF-α levels from EoL-1 cells culture medium in the absence (control) or presence of 10, 20, 30 μM DHA and 500 μM of butyrate by ELISA analysis (*n* = 3, in triplicate). The symbol # denotes statistically significant differences of *p* < 0.05 versus control. *t*-Test analysis for independent samples was applied by Statistica software.

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
