# Peer review of "Docosahexaenoic Acid Inhibits Proliferation of EoL-1 Leukemia Cells and Induces Cell Cycle Arrest and Cell Differentiation"

_nutrients, 2019, doi:10.3390/nu11030574_

Round 1
Reviewer 1 Report
The authors responded appropriately to the requests and the manuscript was improved.
The authors could better explain two points:
-line 501, the authors could cite a paper about the role of PAF / PTAF pathway in inflammation
-line 512 the authors could cite a paper about cytokine production in cell differentiation.
This is to better validate the hypotheses reported
Author Response
The authors could better explain two points:
-line 501, the authors could cite a paper about the role of PAF / PTAF pathway in inflammation
Response:
Line 482, 1 reference was added:
Honda Z1, Ishii S, Shimizu T. Platelet-activating factor receptor. J Biochem. 2002 Jun;131(6):773-9.
-line 512 the authors could cite a paper about cytokine production in cell differentiation.
Response:
Line 492, 2 references were added:
Resnitzky D, Kimchi A. Deregulated c-myc expression abrogates the interferon- and interleukin 6-mediated G0/G1 cell cycle arrest but not other inhibitory responses in M1 myeloblastic cells. Cell Growth Differ. 1991 Jan;2(1):33-41.
Moran DM, Mattocks MA, Cahill PA, Koniaris LG, McKillop IH. Interleukin-6 mediates G(0)/G(1) growth arrest in hepatocellular carcinoma through a STAT 3-dependent pathway. J Surg Res. 2008 Jun 1;147(1):23-33. Epub 2007 Jun 14.
Reviewer 2 Report
In this study, the authors show the effect of docosahexaenoic acid (DHA) on EOL-1 cells. The authors show that DHA in EOL-1 induces cell cycle arrest in G0/G1 phase and cell differentiation. the effect of DHA is associated with alteration of enzymatic activities that regulate PAF levels. The data are clear and well presented.
Author Response
Thank you very much for your comments.
This manuscript is a resubmission of an earlier submission. The following is a list of the peer review reports and author responses from that submission.
Round 1
Reviewer 1 Report
The authors conclude that DHA is a natural substance which could be tested as a future therapeutic tool against F/P+ hypereosinophilic syndrome.
In my opinion this work is well written and sound good but it suffers in many parts and can not be accepted without major changes.
In particular:
1.The authors discussed effects on inflammation of DHA but without any feedback in the manuscript. The authors could performed experiments about inflammation pathway (e.g. qPCR assay and ELISA assay on IL6 and TNFalpha expression after DHA treatment.).
2. The authors study the combined effect of DHA and butyrate but do not provide information on possible addictive or synergistic effects. The authors could performed the Combination Index assay.
See the manuscript: Papi A, Govoni M, Ciavarella C, Spisni E, Orlandi M, Farabegoli F. Epigallocatechin-3-gallate increases RXR-mediated pro-apoptotic and anti-invasive effects in gastrointestinal cancer cell lines. Curr Cancer Drug Targets. 2016;16(4):373-85.
3. The study is performed in a single cell line of leukemia. It could also be performed in a primary leukemia cell line and in normal lymphocytes to confirm the absence of side effects
Minor:
Figure 3 is missing the bar in the images
Author Response
Comments and answers
The authors conclude that DHA is a natural substance which could be tested as a future
therapeutic tool against F/P+ hypereosinophilic syndrome.
In my opinion this work is well written and sound good but it suffers in many parts and can
not be accepted without major changes.
In particular:
1. The authors discussed effects on inflammation of DHA but without any feedback in the
manuscript. The authors could performed experiments about inflammation pathway
(e.g. qPCR assay and ELISA assay on IL6 and TNFalpha expression after DHA treatment.).
Answer: Extra experiments by ELISA assay regarding IL-6 and TNF-α expression were
performed 3 days after DHA treatment. These are depicted in figure 7. An extended paragraph
(paragraph 3.8) in Results and Discussion section of the manuscript was added reporting the
results of cytokines measurement as well as a short discussion on the inflammatory or antiinflammatory
effect of DHA on EOL-1 cells.
2. The authors study the combined effect of DHA and butyrate but do not provide
information on possible addictive or synergistic effects. The authors could performed
the Combination Index assay.
Answer: The Combination Index assay was performed using CompuSyn software and applying
the concentrations used all over the study. Synergistic effect (CI<1) was indicated with DHA
and butyrate. The results were written in Results and Discussion section, paragraph 3.1.
See the manuscript: Papi A, Govoni M, Ciavarella C, Spisni E, Orlandi M, Farabegoli
F. Epigallocatechin-3-gallate increases RXR-mediated pro-apoptotic and antiinvasive
effects in gastrointestinal cancer cell lines. Curr Cancer Drug Targets.
2016;16(4):373-85.
Answer: The above manuscript was cited in the references section of the preset manuscript.
3. The study is performed in a single cell line of leukemia. It could also be performed in a
primary leukemia cell line and in normal lymphocytes to confirm the absence of side
effects
Answer: This would be very interesting and could answer questions such as the side effects in
normal lymphocytes and more. The part of such study could be a future perspective because
the period of time needed for us in order to repeat all the study using in parallel a primary
leukemia cell line and normal lymphocytes, would reach one or two years more. All this work
could not be performed in the short frame of 2-3 months which the journal provides for the
reviewing. Moreover, DHA is a natural substance and was applied in EOL-1 cells in most of the
experiments at very low concentrations, one of the lowest in bibliography, and this is 10 μM.
No other substances (such as various drugs) were used with DHA so as side effects could
appear.
Minor:
Figure 3 is missing the bar in the images
Answer: The magnification (x100) was inserted on the pictures C, D, E.
Reviewer 2 Report
In this study, the authors show the effect of docosahexaenoic acid (DHA) on EOL-1 cells. The authors show that DHA in EOL-1 induces cell cycle arrest in G0/G1 phase and cell differentiation. the effect of DHA is associated with alteration of enzymatic activities that regulate PAF levels.
The data are clear and well presented.
I Have just a few minor comments.
1) In Figure 5A, is shown a time-course of PTAFR mRNA expression levels. I suppose that the used DHA dose is equal to 10 microMolar. Please, specify this in the legend of the figure.
2) In Figure 6 PLA2 is shown that the activity is undetectable in some experimental points. Please, calculate or indicate the cut-off of the sensitivity for the assay used.
3) In the section "Results and discussion" in lines 444 and 445, the description of results recalls figure 4 instead of figure 5.
4)Throughout the manuscript, there is the expression "enzymic activity". Please, use "enzyme activity" or "enzymatic activity"
Author Response
Comments and answers
In this study, the authors show the effect of docosahexaenoic acid (DHA) on EOL-1
cells. The authors show that DHA in EOL-1 induces cell cycle arrest in G0/G1 phase and
cell differentiation. the effect of DHA is associated with alteration of enzymatic
activities that regulate PAF levels.
The data are clear and well presented.
I Have just a few minor comments.
1) In Figure 5A, is shown a time-course of PTAFR mRNA expression levels. I
suppose that the used DHA dose is equal to 10 microMolar. Please, specify this in
the legend of the figure.
Answer: 10 μΜ DHA was used. The specification was inserted in the figure legend.
2) In Figure 6 PLA2 is shown that the activity is undetectable in some experimental
points. Please, calculate or indicate the cut-off of the sensitivity for the assay used.
Answer: The phrase “For PLA2 and PAFAH activities, the sensitivity of the assay used was 0,1 pmoles
C12- or C6-NBD-FA/h/μg protein” was inserted in paragraph “Total PLA2 and PAFAH activities in EOL-1”
in Materials and Methods section. The phrase “PLA2 activity was not detectable in culture medium at
10, 30 μM DHA and in butyrate” was inserted in the legend of figure 6.
3) In the section "Results and discussion" in lines 444 and 445, the description of
results recalls figure 4 instead of figure 5.
Answer: It was corrected in the new paragraph 3.9.
4) Throughout the manuscript, there is the expression "enzymic activity". Please,
use "enzyme activity" or "enzymatic activity"
Answer: It was corrected.